# Transcriptomic Alterations Induced by Tetrahydrocannabinol in SIV/HIV Infection: A Systematic Review

**DOI:** 10.3390/ijms26062598

**Published:** 2025-03-13

**Authors:** Amir Valizadeh, Rebecca T. Veenhuis, Brooklyn A. Bradley, Ke Xu

**Affiliations:** 1Department of Psychiatry, Yale School of Medicine, New Haven, CT 06510, USA; amir.valizadeh@yale.edu (A.V.); brooklyn.bradley@cox.net (B.A.B.); 2VA Connecticut Healthcare System, West Haven, CT 06516, USA; 3Department of Molecular and Comparative Pathobiology and Neurology, Johns Hopkins School of Medicine, Baltimore, MD 21287, USA; rterill1@jhmi.edu

**Keywords:** cannabinoids, gene expression, gene expression regulation, human immunodeficiency virus, simian immunodeficiency virus, acquired immunodeficiency syndrome

## Abstract

Given the high prevalence of cannabis use among people with HIV (PWH) and its potential to modulate immune responses and reduce inflammation, this systematic review examines preclinical evidence on how tetrahydrocannabinol (THC), a key compound in cannabis, affects gene and micro-RNA expression in simian immunodeficiency virus (SIV)-infected macaques and HIV-infected human cells. Through a comprehensive search, 19 studies were identified, primarily involving SIV-infected macaques, with a pooled sample size of 176, though methodological quality varied across the studies. Pathway analysis of differentially expressed genes (DEGs) and miRNAs associated with THC revealed enrichment in pathways related to inflammation, epithelial cell proliferation, and adhesion. Notably, some DEGs were targets of the differentially expressed miRNAs, suggesting that epigenetic regulation may contribute to THC’s effects on gene function. These findings indicate that THC may help mitigate chronic immune activation in HIV infection by altering gene and miRNA expression, suggesting its potential immunomodulatory role. However, the evidence is constrained by small sample sizes and inconsistencies across studies. Further research employing advanced methodologies and larger cohorts is essential to confirm THC’s potential as a complementary therapy for PWH and fully elucidate the underlying mechanisms, which could inform targeted interventions to harness its immunomodulatory effects.

## 1. Introduction

### 1.1. Rationale

The emergence of human immunodeficiency virus (HIV) in the 1920s and its spread across Europe and North America in the 1980s marked the onset of a persistent global health challenge. As of 2020, an estimated 37.7 million cases have been reported, with 1.5 million new cases and 680,000 deaths attributed to acquired immunodeficiency syndrome (AIDS)-related causes worldwide [1]. During the era of antiretroviral therapy (ART), which has been remarkably successful in suppressing HIV to undetectable levels in plasma and preventing transmission, a significant proportion of people with HIV (PWH) still experience non-AIDS comorbidities. These conditions, such as atherosclerosis, liver fibrosis, and neuroinflammation, arise from sustained chronic immune activation and inflammation [2], which may be driven by adverse effects of either living with HIV or the ART itself—factors that are often difficult to disentangle [3]. Studies in natural hosts of simian immunodeficiency virus (SIV, a virus closely associated with HIV-1), such as sooty mangabeys and African green monkeys, have demonstrated that high viral replication without chronic inflammation does not lead to AIDS progression [4,5]. In contrast, non-natural hosts, including rhesus macaques and pig-tailed macaques infected with SIV or simian–human immunodeficiency virus (SHIV), as well as some chimpanzees infected with HIV-1, exhibit progression to AIDS, which is associated with both high viral replication and significant chronic inflammation [6,7]. These observations suggest that chronic inflammation may be a critical factor contributing to disease progression in pathogenic lentiviral infections and potentially offers insights into HIV-1 pathogenesis in humans. Although ART remains the cornerstone of treatment for PWH and effectively halts progression to AIDS, adjunctive use of immunomodulatory agents such as cannabis may help relieve symptoms or reduce inflammation [8].

Cannabis consumption has emerged as the most prevalent drug use among PWH. A national survey in 2022 reported that 74.6% of PWH had lifetime marijuana use, compared with 54.2% of people without HIV (PWoH). Similarly, marijuana use within the previous 12 months was also significantly higher among PWH compared with PWoH (44.9% versus 24.4%), as was marijuana use within the previous month (35.6% versus 14.2%) [9]. In 1985, the Food and Drug Administration (FDA) approved dronabinol, a synthetic form of delta-9-tetrahydrocannabinol (Δ-9-THC, hereafter referred to as simply THC) for managing HIV-associated anorexia [10,11]. Additionally, the potential of cannabis as an adjunctive therapeutic agent to ART for HIV infection has recently garnered significant scientific interest.

Cannabinoids (CBs) are a class of compounds found in the cannabis plant. The endocannabinoid system encompasses an extensive network of naturally occurring CBs (endocannabinoids) and cannabinoid receptors (CBRs) widely distributed throughout the brain and body [12]. Two CBRs have been characterized: CB receptor 1 (CBR1) and CBR2. CBR1, while also located in other organs, is primarily distributed in the central nervous system (CNS) [13]. However, CBR2 is predominantly found in immune cells, particularly in B lymphocytes, natural killer cells, macrophages, polymorphonuclear cells, CD4^+^ T cells, and CD8^+^ T cells [14]. The most prominent exocannabinoids found in the cannabis plant, *Cannabis sativa*, are THC and cannabidiol (CBD). THC, the primary psychotropic component, acts as a partial agonist of both CBR1 and CBR2 and is recognized as an immunomodulator [15,16,17,18]. In contrast, CBD does not exhibit psychotropic effects and is known for its significant antioxidant properties. Previous research suggests that THC has a strong anti-inflammatory effect, as it has been shown to decrease mortality associated with acute respiratory distress syndrome [19], suppress experimental autoimmune encephalomyelitis (a laboratory model of multiple sclerosis) [20], and mitigate inflammation in rheumatoid arthritis [21] due to its role in modulating the activity of macrophages, T cells, and inflammatory cytokines [22,23,24].

A previous review outlined the immunomodulatory changes induced by cannabis in PWH [25]. However, the mechanisms and efficacy of cannabinoids in ameliorating inflammation-associated pathologies, especially among PWH, remain to be definitively elucidated. In recent years, there has been an increase in research investigating the impact of THC on gene expression within host genomes infected with HIV. Most of these studies have been conducted using Asian macaques infected with SIV. These investigations provide valuable insights into how THC alters the transcriptome of HIV-infected cells and the potential role of cannabis in the well-being of PWH.

### 1.2. Objectives

This systematic review comprehensively assesses the available preclinical studies, as well as in vitro studies on human cells, to elucidate the potential mechanisms by which cannabis may mitigate HIV symptoms through modulation of the host transcriptome in response to HIV-1 infection.

## 2. Methods

The design and methodology of this review were completed following the Center for Reviews and Dissemination (CRD) Guidance for Undertaking Reviews in Healthcare [26]. This guidance advocates for robust and methodical approaches to conducting systematic reviews. This review aligns with the Preferred Reporting Items for Systematic Reviews and Meta-Analyses (PRISMA) guidelines [27], ensuring the comprehensive and transparent documentation of our systematic review process.

### 2.1. Eligibility Criteria

Much of our understanding of the immunomodulatory effects of cannabis is derived from studies involving SIV-infected macaques [28]. The SIV virus has been extensively utilized to investigate HIV and AIDS in animal models [29]. Our preliminary search revealed that most of the relevant studies focused on in vivo investigations of SIV-infected rhesus macaques. Therefore, we opted to prioritize this population for our review. While we also aimed to include studies involving humans or human cells, only those involving rhesus macaques were included in our subsequent analyses. Also, given the diverse and potentially contrasting effects of cannabinoids mediated through different CBRs, we opted to narrow our focus to THC, the most extensively studied cannabinoid in the literature.

The inclusion and exclusion criteria were informed using the population, intervention, comparator, and outcome (PICO) framework. The inclusion criteria were as follows: (1) Population: studies involving macaque models, humans, and human cells; (2) Intervention: given that the majority of published papers focused on THC, in the current review, we included only studies using THC; (3) Comparator: studies using vehicle, placebo, or no treatment as the control group; and (4) Outcomes: (a) Primary: studies examining alterations in gene expression (mRNA) or micro-RNA (miRNA) expression in the host transcriptome; and (b) Secondary: studies examining immune cell markers and HIV/SIV viral load to elucidate whether THC changed the proportions of immune cell types or the replication of the virus, representing intriguing aspects of the research on the interplay between THC, immune cell population, and viral replication in cases of HIV.

Exclusion criteria included the following: (1) studies using other cannabinoids or a combination of THC and another cannabinoid were excluded to reduce the risk of confounding bias; (2) studies that lacked a control group (vehicle, placebo, or no treatment); and (3) studies that did not assess gene/miRNA expression, immune markers, or HIV/SIV viral load, or solely involved non-relevant animal models (e.g., rodents) without any human cell cultures.

### 2.2. Information Sources

We searched eligible publications from four databases up to February 2025, including APA PsycInfo (through Ovid), Embase (through Ovid), MEDLINE (through Ovid), and PubMed. Additionally, the bibliographies of relevant studies were examined to identify additional pertinent data. Only studies published in English were considered for inclusion.

### 2.3. Search Strategy

The search strategy is presented in Appendix A.

### 2.4. Selection Process

Two reviewers independently screened the retrieved records for inclusion. Discrepancies were resolved through discussion and consensus.

### 2.5. Data Collection Process

A data extraction form was designed and used to gather information from the included studies. Two reviewers extracted the data independently and cross-checked their results to ensure accuracy. A third reviewer was consulted in cases of discrepancies. If data were missing, the study authors were contacted.

### 2.6. Data Items

The following information was extracted from the included studies: (1) study ID, publication year, country, study design, the number of groups, and animals per group; (2) sample size and characteristics (age, sex, weight, ethnicity, coexisting conditions); (3) intervention details (dosage, duration); (4) infection details (virus strain, infection timing); (5) timing of THC administration and follow-up duration; (6) methods for assessing gene and miRNA expression and statistical analysis; and (7) measurements of viral load and immune cell markers.

### 2.7. Study Risk of Bias Assessment

Two reviewers independently evaluated the methodological and reporting quality of the included studies using a modified version of the Q-Genie tool [30], with discrepancies resolved by a third reviewer. The Q-Genie comprises 11 items addressing biases such as the scientific basis for the research question, comparison group ascertainment, classification of genetic variants and outcomes, bias sources, sample size adequacy, statistical analysis plans, testing of assumptions (e.g., Hardy–Weinberg equilibrium), and interpretation of results. Each item was rated on a scale from 1 (poor) to 7 (excellent).

Since the focus of our review differed from the original scope of the Q-Genie tool, we adapted it to better suit the assessment of the included studies. Specifically, we revised item 2 to address technical aspects related to measuring the outcome, and item 4 was modified to include technical measures in sample recruitment, to minimize potential confounding effects. Additionally, item 5 was updated to evaluate technical measures for implementing the intervention. We excluded item 7 to avoid subjective judgments regarding sample size; instead, this information is presented separately for each study. The modified version of the tool, along with our criteria for rating each item, is detailed in Appendix A. More specifically, this modified version evaluates the reporting quality, study methods, and potential presence of the following biases: confirmation bias (Q1 and Q10), measurement bias (Q2), selection bias (Q3), confounding bias (Q4), implementation bias (Q5), reporting bias (Q6), analytical bias (Q7), statistical bias (Q8), and modeling bias (Q9). We refrained from calculating total scores to avoid potentially misleading interpretations that could arise from combining each study’s weaknesses and strengths into a single score.

### 2.8. Effect Measures

We compared the expression levels of differentially expressed genes (DEGs) and miRNAs that were either significantly upregulated or significantly downregulated between the case and control groups. A similar method was used for the cell surface markers. For assessment of viral load, the reported levels for each subject were used directly for the analysis.

### 2.9. Synthesis Methods

DEGs and miRNAs associated with THC were extracted. The target genes of the differentially expressed miRNAs were obtained from miRNet [31]. We also extracted viral load data for each study on a per-subject basis to avoid the duplication of subjects, and we conducted *t*-tests to compare viral load levels between the groups.

Analyzing miRNA enrichment across different species presents several challenges, such as inconsistent miRNA annotations and limitations in merging cross-species data. For example, annotations in rhesus macaque (*Macaca mulatta*) may not align perfectly with those of humans (*Homo sapiens*). To address this issue, we employed orthologous miRNA substitutions, a method that has previously been utilized in the literature. For instance, Ganie et al. [32] analyzed miRNA genes across ten rice (*Oryza sativa*) species, identifying orthologous miRNAs based on sequence conservation, to study evolutionary changes. Similarly, Sadanandam et al. [33] compared mRNA and miRNA transcriptomes in pancreatic neuroendocrine tumors between mice and humans, using sequence similarity to classify subtypes. Applying a similar approach, only those differentially expressed (DE) miRNAs that were deemed conserved across species were included in the current analyses. miRNA sequences were extracted from miRBase [34], a database of published miRNA sequences and annotations.

We utilized Gene Ontology Biological Process (GO BP) terms as the primary database for the functional enrichment analyses. We conducted separate analyses of the DEGs and target genes of the DE miRNAs. These analyses were carried out using enrichR [35]. To summarize the lists of GO terms, we utilized Revigo [36]. The significance threshold for all analyses was set to 0.05 following correction of the false discovery rate (FDR).

## 3. Results

### 3.1. Study Selection

We identified 66 records, 60 from databases and 6 from citation searching, from which 20 reports of 19 studies [37,38,39,40,41,42,43,44,45,46,47,48,49,50,51,52,53,54,55,56] were included in this review (16 on SIV-infected macaques and 3 on human cells). One report [50] contained only viral load data that had already been reported in other studies. Figure 1 depicts the study selection process.

### 3.2. Study Characteristics

A summary of the characteristics of the studies included in this review is presented in Figure 2.

Except for Wei et al., 2016 [52], which used Chinese macaques (*Macaca fascicularis*), all other 15 macaque studies used Indian macaques (*Macaca mulatta*). All macaque studies used male subjects except for Amedee et al., 2014 [38], which used female macaques; Premadasa et al., 2023 [49], which used a combination of both male and female macaques; and Kumar et al., 2016 [42], which did not report the sex of the subjects. The pooled sample size of the 16 studies on macaques was 176, with a median of 8 (range: 4–25). The median duration of follow-up for those studies was 210 days (range: 90–540). None of the studies included samples receiving ART in their outcomes of interest. One of the subjects in two of the studies [37,43] was infected with SIVmac239, while the other subjects were infected with SIVmac251. Another study [44] used SIVmac251-infected subjects in the treatment group and SIVmac239-infected subjects in the control group. Also, Kumar et al., 2016 [42] did not report the strain of the virus. Other studies solely used SIVmac251-infected subjects. Both SIVmac251 and SIVmac239 are known to be CCR5-tropic and were originally isolated from the same macaque [57,58]. The other three non-macaque studies used HIV-1 MN [54], HIV-1 NL4-3 [55], and HIV-1 AD [56] strains, respectively.

### 3.3. Risk of Bias in Studies

A summary of the methodological and reporting quality of the studies included in this review is presented in Figure 3.

The fifth item of the quality ratings (the technical measures used to implement the intervention) had the highest mean across the studies (6.7), indicating that the dose and route of administration were reported and justified in almost all the included studies. The items with the next highest quality ratings were the ninth item (description and test of all assumptions; 5.8) and the first item (adequacy of the presented hypothesis and rationale; 6.1), while the sixth item (disclosure and discussion of sources of bias) had the lowest quality rating (4.2), indicating that the included studies lacked thorough reporting and discussion of the possible sources. Thirteen studies had a mean quality score of 5 or more, while the remaining studies (n = 6) had scores below that cutoff.

### 3.4. Results of Individual Studies and Syntheses

#### 3.4.1. THC Alters Gene Expression in SIV-Infected Macaques

Out of the sixteen studies on macaques, eight included differential gene expression analysis. The pooled sample size for the subjects included in gene expression evaluation in these studies was 94, with a median of 10 (range: 4–17). The median duration of follow-up for those studies was 210 days (range: 90–540). Three studies used the RNA sequencing (RNA-Seq) technique, three used microarray assays, one used quantitative real-time polymerase chain reaction (qRT–PCR), and one used both gene expression microarray assays and qRT–PCR validation.

A total of 294 DEGs were reported (*p* < 0.05), of which 109 were significantly upregulated and 185 were significantly downregulated in SIV+/THC+ subjects compared with SIV+/THC− controls. A total of 7 of the 294 DEGs were reported to be downregulated in THC-treated subjects in more than one study, as follows:*CSF3R* (a cytokine that controls the production, differentiation, and function of granulocytes [59]) in the colon [43] and gingiva [44];*DEFA4* (a pro-inflammatory alpha defensin gene [60]) in the colon [42,43];*DEFA5* and *DEFA6* (also pro-inflammatory alpha defensin genes [60]) in the colon [42] and colonic epithelium [50];*KLK6* (a member of the kallikrein subfamily of the peptidase S1 family of serine proteases [61]) in the oropharyngeal mucosa [37] and gingiva [44];*MMP8* (a modulator of the activity of pro-inflammatory cytokines such as TNF-α and IL-1β [62]) in the colon [42,43];*SOCS3* (a cytokine-inducible negative regulator of cytokine signaling [63]) in the oropharyngeal mucosa [37] and colon [43].

Also, the following gene was reported as upregulated in more than one study:*PAPPA* (a protein that plays a role in bone formation, inflammation, and wound healing [64]) in the oropharyngeal mucosa [37] and gingiva [44].

THC downregulated multiple pro-inflammatory genes as well as genes involved in the negative regulation of inflammation, such as *SOCS3*, *NR1D1*, and *MRC1*, all of which are typically induced by inflammatory stimuli to help modulate or restrain the immune response [65,66,67]. Because these regulatory genes are upregulated in the presence of inflammation, their reduced expression under THC treatment suggests that the overall inflammatory stimulus was diminished, thereby lowering the need for compensatory anti-inflammatory mechanisms. In contrast, THC upregulated genes that actively promote an anti-inflammatory environment, including *TGFB2*, *PARD3B*, and *TRIM35*. *TGFB2*, for example, is a potent anti-inflammatory cytokine [68], and its increased expression is consistent with enhanced suppression of immune activation and promotion of regulatory T-cell activity. Together, these findings highlight THC’s capacity to actively reinforce anti-inflammatory pathways, complementing the reduced need for regulatory feedback observed with *SOCS3*, *NR1D1*, and *MRC1*. THC also upregulated genes critical for tissue protection and resolution of inflammation, including *HP*, *PDGFC*, and *TIMP2*. *HP* (haptoglobin), which binds free hemoglobin to mitigate oxidative stress [69], showed increased expression, indicating a protective role against inflammation-induced tissue damage. *PDGFC*, a growth factor promoting angiogenesis and tissue regeneration [70], was elevated, underscoring THC’s support for post-inflammatory repair. *TIMP2*, an inhibitor of matrix metalloproteinases [71], exhibited higher levels, suggesting reduced degradation of the extracellular matrix and enhanced tissue integrity during inflammation. The differential regulation of these genes by THC reveals a dual mechanism of action: it simultaneously reduces the inflammatory stimulus and actively promotes anti-inflammatory and reparative processes. Table 1 outlines several DEGs identified in the included studies, focusing on those relevant to HIV-associated inflammatory complications.

Although THC did not exert inhibitory effects on certain other pro-inflammatory genes, such as *TNF*, *IL1B*, and *CCL2*, in the study by Chandra et al., 2015 [39], it should be noted that that study was conducted in the acute phase of SIV infection (60 days post-infection). Interestingly, the study by Simon et al. in 2016 on the striatum of the brain [51], although conducted in the acute phase of SIV infection, found decreased *TNF* expression levels in THC-treated subjects compared with controls. These data suggest that the effects of THC on pro-inflammatory genes may be tissue- and timing-specific, whereas the induction of anti-inflammatory genes appears to be consistent across multiple tissues.

Our pathway analysis revealed that 294 DEGs were enriched in 41 significant GO BP pathways (FDR < 0.05). The list of significant GO BP terms for the DEGs is presented in Table 2. All genes are presented in Appendix A, and the pathways are presented in Appendix A.

#### 3.4.2. THC Alters miRNA Expression in SIV-Infected Macaques

Out of the 16 studies on macaques, 8 evaluated differential miRNA expression analysis. The pooled sample size for these studies was 80, with a median of 8 (range: 2–17). The median duration of follow-up for those studies was 210 days (range: 90–540). Six studies used miRNA microarray analysis, and three used the small RNA-Seq (sRNA-Seq) technique. Four of these nine studies validated the results using qRT-PCR.

In these studies, 143 DE miRNAs were reported, 79 of which were significantly upregulated and 55 of which were significantly downregulated in SIV+/THC+ subjects compared with SIV+/THC− controls. Conflicting results were also obtained for nine miRNAs that were reported as upregulated in one study and downregulated in another. Also, 10 miRNAs (excluding those with conflicting results) were found to be differentially expressed in more than one study. Five of those ten were upregulated in at least two different studies, as follows:miR-105-5p (associated with nervous system development [51]), in extracellular vesicles (EVs) from the basal ganglia [40] and the striatum of the brain [51];miR-193b-5p (involved in tumor suppression and modulating inflammatory responses [72]) in the colon [42] and gingiva [44];miR-99a-5p (associated with the EGFR tyrosine kinase inhibitor resistance pathway [41]) in EVs and extracellular condensates (ECs) of plasma [41] and in the gingiva [44];miR-99b-5p (an immune response modulator and tumor suppressor [73]) in the duodenum [39] and gingiva [44];miR-374a-5p (associated with inflammation regulation [74]) in EVs from the basal ganglia [39] and colon [42].

Meanwhile, the five miRNAs that were downregulated in more than one study included the following:miR-19b-1-5p (involved in the DNA damage response [64]) in ECs from plasma [41] and in the colon [43];miR-21-3p (associated with the immune response in sepsis [64]) also in ECs from plasma [41] and the colon [43];miR-223-3p (associated with hereditary neutrophilia [64]) in the gingiva [44] and CD4 cells [46];miR-382-3p (associated with cell apoptosis [41]) in EVs from plasma [41] and the gingiva [44];miR-502-3p (a modulator of glutamatergic and GABAergic synapse function [75]) in the gingiva [44] and basal ganglia [45].

Table 3 outlines several DE miRNAs identified in the included studies, focusing on those relevant to HIV-associated inflammatory complications.

These studies delineated the immunomodulatory function of THC across various tissue types, including the duodenum, extracellular vesicles within the basal ganglia and plasma, the colon, gingiva, and the basal ganglia and striatum of the brain, in SIV+/THC+ macaques compared with SIV+/THC− controls.

We further assessed whether the DE miRNAs regulated any of the DEGs found in our review. We found that 17 genes were among the DEGs found in the included studies and also among the potential target genes (identified using miRNet) of DE miRNAs reported in the included studies (*ITGA3*, *KATNAL1*, *NF1*, *OLFML2A*, *CLIC5*, *BMPR2*, *TNF*, *KRT80*, *EGFR*, *SLC6A4*, *SLC7A2*, *CEBPG*, *TGFB2*, *SLC16A1*, *ALDH9A1*, *PAN2*, and *CXCL12*). Some of these genes are involved in diverse cellular functions that collectively contribute to immune cell activation, cytokine production, and inflammatory signaling cascades. For instance, *TNF* and *TGFB2* are well-established mediators of inflammation, modulating immune cell responses and regulating tissue homeostasis [76,77,78]. Furthermore, *TGA3* and *EGFR* are potentially involved in cell adhesion, migration, and proliferation, which are crucial for immune cell trafficking and inflammatory cell recruitment to sites of inflammation [79,80]. Additionally, *CXCL12* is a key chemokine involved in inflammation and immune regulation. It guides immune cell migration to sites of inflammation by binding to its receptor, *CXCR4*, and regulates hematopoiesis and tissue repair [81]. It also modulates immune cell function, and its dysregulation is linked to inflammatory diseases and autoimmune disorders [82]. Figure 4 presents the top five DE miRNAs with the highest numbers of connections with genes that appeared on both the list of DEGs and the list of target genes of DE miRNAs.

Following orthologous miRNA replacement, a total of 121 miRNAs were used for the pathway analyses. The top 25 significant GO BP terms (ordered by the *p* values) for the differentially expressed miRNAs are highlighted in Figure 5. All significant pathways are presented in Appendix A.

#### 3.4.3. The Impact of THC on Immune Cell Markers and Viral Load in SIV

In addition to the reported alterations in gene and miRNA expression, we noted that six of the studies in macaques included fluorescence-activated cell sorting (FACS) analysis. A total of 20 immune cell markers were evaluated, 17 of which were associated with T cells, 2 with B cells, and 1 with macrophages. It should be noted that the only study that evaluated B-cell markers [52] was conducted in Chinese macaques, which are known to be more resistant to SIV infection and have lower inflammatory responses to SIV infection compared with Indian macaques [83,84]. In addition, one study was completed using only female macaques [38] and reported an increase in the population of CD4^+^ central (CD95^+^CD28^+^) and effector (CD95^+^CD28^−^) cells in the duodenum of THC+ compared with THC− subjects, while these markers were not differentially expressed in a similar study using male macaques [48]. However, neither of these studies reported a difference in these cells in the peripheral blood of THC+ compared with THC− subjects. The markers reported in the included studies and their relative levels in SIV+/THC+ compared with SIV+/THC− macaques are presented in Table 4 and Appendix A.

Additionally, to evaluate whether THC affects the replication of the virus, we also evaluated the viral load in SIV+/THC+ compared with SIV+/THC− macaques. Twelve studies evaluated viral load alterations. The pooled sample size of these studies was 70, with a median of 8 (range: 6–25). The target tissues included the brain, colon, cerebrospinal fluid, duodenum, gingiva, oropharyngeal mucosa, and plasma. We pooled the data for each target tissue separately and performed *t*-tests to compare the means between the groups. The results are provided in Appendix A.

Our results indicate that THC administration did not significantly change the viral load in any of the target tissues, although the sample size for most of the comparisons was small and those tests lacked power. We also performed pooled sample analysis by pooling data from cases and comparing them with the pooled data of the controls. The SIV+/THC+ cases had a mean viral load of 915.9 × 10^6^/mL (SD = 5048.9, N = 67), while the SIV+/THC− controls had a mean viral load of 189.4 × 10^6^/mL (SD = 641.1, N = 80). This difference was not deemed statistically significant (*p* = 0.21).

#### 3.4.4. Studies on HIV-Infected Human Cells

Only three studies in HIV-infected human cell models have examined THC-induced alterations and reported outcomes eligible for this review [54,55,56]. All these studies were conducted in short timeframes (3–19 days). Overall, these studies had relatively low methodological and reporting quality scores (mean quality scores of 3 to 5.6). All investigations were conducted using healthy human cell cultures subjected to incubation with the HIV-1 virus within laboratory settings. One of these studies was an ex vivo study utilizing human cell implants in a murine model [55], while the other two were in vitro studies on human cells [54,56].

Noe et al. [54] evaluated the effect of THC administration on syncytium formation (as an indication of virus infection and cytopathicity) in HIV-infected MT-2 cells. MT-2 cells are a human T-cell line infected with HTLV-1, derived from male umbilical cord lymphocytes co-cultured with leukemic cells from an adult T-cell leukemia patient. They act like regulatory T cells, lack CD3 expression, and are used to study HIV replication and immune suppression [85]. A significant increase in syncytium formation was found in cells treated with THC at different concentrations (0.5, 1, 3, and 5 µg/mL) compared with control vehicle-treated cells after 3 days. The authors concluded that THC may enhance cell-free HIV-1 infection of MT-2 cells.

Roth et al. [55] evaluated the effects of THC on human peripheral blood leukocytes implanted into severe combined immunodeficient (SCID) mice (huPBL-SCID). THC administration at an intraperitoneal concentration of 10 mg/kg was started 5 days after HIV infection. After 5–7 days of THC administration, the cells were harvested via intraperitoneal lavage, and the expression of different immune cell markers was evaluated. A significant increase in the CD45^+^CCR5^+^ cell population was observed following 5 days of THC treatment, returning to normal after another 5 days. No significant changes were observed in the CD45^+^CXCR4^+^ cell population. The authors also reported a 50-fold increase in HIV RNA copy number in the peripheral blood of HIV+ THC-treated mice compared with HIV+ VEH-treated controls. They concluded that exposure to THC can increase HIV coreceptor expression (CCR5 only, not CXCR4) and act as a cofactor to significantly enhance HIV replication.

Finally, Williams et al. [56] evaluated the effects of THC administration on HIV-infected human monocyte-derived macrophages (MDMs). They found a significant decrease in CXCR4^+^, CCR5^+^, CD4^+^, CD14^+^, CD16^+^, and CD163^+^ cell populations, while no significant changes in the CD11b^+^, ICAM-1^+^, or Mac-387^+^ populations were reported. The authors concluded that the mechanism by which THC suppresses HIV-1 infection includes a reduction in HIV receptor (CD4, CCR5, and CXCR4) expression on the cell surface, which diminishes entry efficiency.

## 4. Discussion

### 4.1. Interpretation of Results

Despite the high prevalence of cannabis use among PWH, little is known about how cannabinoids alter gene and miRNA expression at the transcriptome level. This systematic review assesses studies involving SIV-infected macaques to shed light on how THC influences the expression of genes and miRNAs related to immune and inflammatory functions.

The notable changes in gene expression induced by THC suggest its anti-inflammatory properties in peripheral tissues, including the oropharyngeal mucosa, gingiva, and colon. These studies highlight THC’s ability to modulate the integrity of oral and intestinal epithelial barriers and regulate the inflammatory processes in these tissues, consistent with evidence demonstrating lower levels of markers associated with cell death in intestinal CD4^+^ and CD8^+^ T cells following THC treatment, as well as a reduction in colonic CD8^+^ T-cell activation by THC [43]. The latter may be a product of decreased gut microbiome dysbiosis following THC treatment, which reduces the translocation of microbial products and thus reduces CD8^+^ T-cell activation. In the CNS, THC appears to exhibit an anti-inflammatory effect by reducing the expression of pro-inflammatory cytokine genes and potentially enhancing neuroprotection through increased *BDNF* expression, which may promote neural survival [86,87]. Overall, THC’s immunomodulatory effects appear to vary depending on the tissue involved. This conclusion becomes even more likely when we consider that most of the included studies were conducted on similar subjects and used the same methods. A previous study [88] indeed showed that THC inhibits HIV’s Tat-enhanced monocyte adhesion to extracellular matrix proteins via CBR2 activation linked to altered β1-integrin expression and actin polymerization, suggesting THC may reduce immune cells’ infiltration into some tissues, further supporting its tissue-specific anti-inflammatory role.

Additionally, we observed that the genes affected by THC were enriched in various pathways related to immune responses and the regulation of inflammation. The regulation of PI3K signaling, for example, is crucial for modulating various cellular processes, including immune cell activation and cytokine production [89]. Similarly, the JAK-STAT signaling pathway plays a pivotal role in mediating responses to cytokines and growth factors [90], thereby influencing immune cell function and inflammation. Its positive regulation of the reactive oxygen species biosynthetic process suggests involvement in oxidative stress, which is often associated with inflammatory responses [91]. Furthermore, pathways such as the cellular response to lipopolysaccharides and the regulation of the inflammatory response relate directly to the immune system’s role in detecting and responding to microbial pathogens or inflammatory stimuli. These findings underscore the intricate network of pathways involved in immune regulation and inflammation targeted by THC. A previous study [92] also demonstrated that THC suppressed IFNα secretion by plasmacytoid dendritic cells (pDCs) from both healthy and HIV+ donors through impaired phosphorylation of *IRF7*, indicating that THC dampens early antiviral responses, which may influence immune activation in HIV/SIV. Further supporting THC’s immunomodulatory role, Lyu et al. [93] found that THC treatment decreased blood extracellular vesicle (BEV) concentration in SIV-infected rhesus macaques, particularly at 30 days post-infection, and altered their protein cargo, such as tetraspanins CD9 and CD81. Functionally, BEVs from THC-treated macaques (THC+/SIV+ BEVs) reduced the spread of monocytes and their adhesion to collagen, modified cytoskeletal dynamics, and induced distinct signaling pathways (e.g., downregulation of pERK1/2 and upregulation of pFAK and integrin β1) compared with VEH+/SIV+ BEVs. These studies provide further evidence of THC’s involvement in immune regulation and inflammatory functions.

In SIV-infected macaques, THC treatment is associated with changes in gene expression and pathway enrichment that appear to be linked to its activation of the CBR2 receptor, which is predominantly expressed in immune cells [43,47]. Activation of CBR2 by THC triggers intracellular signaling via Gi/o proteins, leading to the inhibition of adenylyl cyclase and a reduction in cyclic AMP (cAMP) levels, which in turn modulates transcription factor activity [94]. In addition, CBR2 engagement activates mitogen-activated protein kinase (MAPK) pathways, including ERK, JNK, and p38, that play roles in controlling inflammatory and immune responses [95,96,97]. Moreover, a previous study [98] reported that THC and the CBR2 agonist JWH-015 reduced HIV p24 levels and altered cytokine production in HIV-infected monocyte-derived dendritic cells (MDDCs), suggesting that THC may directly suppress viral replication and modulate immune function via CBR2 activation. For instance, in one of the included studies, THC was shown to suppress TNF-α production in the striatum during acute SIV infection [51], probably through post-transcriptional mechanisms; one hypothesis posits that MAPK-mediated inhibition of TNF-α converting enzyme (TACE) might be involved [99,100,101]. Regarding HIV, studies with human cell models suggest that THC may indirectly reduce viral transcription by dampening immune activation and influencing transcription factors such as NF-κB, though these effects require further validation [56,102].

While this systematic review centers on preclinical studies in SIV-infected macaques, assessing the applicability of these findings to PWH is critical for understanding THC’s therapeutic potential in managing HIV-associated inflammation. In SIV-infected macaques, THC treatment downregulated pro-inflammatory genes (e.g., *TNF*, *DEFA4*, *MMP8*) and was linked to CBR2 receptor-mediated signaling, suggesting an anti-inflammatory effect consistent with our earlier observations in peripheral tissues and the CNS. These findings resonate with human studies on cannabis use in PWH. For example, Costiniuk and Jenabian [25] reported that cannabinoids, including THC, may attenuate systemic inflammation and immune activation in PWH by modulating cytokine production and T-cell function. Similarly, Watson et al. [8] observed that daily cannabis use correlated with reduced central nervous system inflammation, hinting at neuroprotective benefits. However, human studies also reveal inconsistencies, with some suggesting that cannabis may not uniformly lower inflammatory markers and could, in certain contexts, exacerbate immune dysfunction, potentially due to variations in dosage, administration, or patient profiles. These discrepancies are further complicated by limitations in the macaque model, including differences between SIV and HIV disease progression and the lack of ART in most studies reviewed, unlike the typical clinical scenario for PWH receiving suppressive ART. Previous related studies have also shown the challenges in translating animal findings to humans. For example, a study in 2018 [103] found that T cells from HIV+ donors were less sensitive to THC-mediated suppression of IFNα- and IL-7-induced stimulation compared with those from healthy donors, while another study [104] showed that HIV transgenic rats were less affected by THC’s reduction of prepulse inhibition (PPI), a measure of fronto-striatal function, highlighting variable responses to THC in HIV contexts and animal models. Thus, while our results highlight THC’s potential to mitigate inflammation in lentiviral infections, translating these insights to humans warrants caution and necessitates further clinical investigation.

Building on these insights, we found that DE miRNAs affected by THC treatment target pathways that are essential for immune response and homeostasis. Specifically, the regulation of cysteine-type endopeptidase activity in apoptosis suggests THC’s role in cell survival and death, which is key to regulating populations of immune cells and resolving inflammation [105]. Additionally, THC may influence immune cell function and activation states via pathways such as the transmembrane receptor protein tyrosine kinase signaling pathway and the MAPK cascade [106,107]. THC’s involvement in cellular response to hypoxia and DNA damage implicates it in modulating immune responses under cellular stress and injury, typical in inflammatory environments. Furthermore, THC’s effects on Wnt signaling, autophagy, and cell adhesion highlight its impact on immune cells’ differentiation, migration, and microenvironmental interaction [108]. THC also influences pathways related to cytokine production, macrophage activation, and neuroinflammatory responses, indicating a broad impact on inflammatory processes beyond the immune system.

These findings suggest that THC regulates immune responses and inflammation through miRNA expression. For instance, miRNAs target genes encoding cytokines, receptors, and signaling molecules within the TNF and TGF-β pathways, affecting immune cell activation and cytokine production [109,110]. Additionally, these miRNAs regulate genes involved in cell adhesion, migration, and proliferation, such as *ITGA3*, *EGFR*, and *CXCL12*, impacting immune cell trafficking, recruitment, and tissue repair [111,112,113]. These genes play crucial roles in immune function and inflammation processes. Being both differentially expressed and targeted by DE miRNAs, these genes probably represent key nodes in THC’s immunomodulatory effects, illustrating the complex interplay between THC, gene expression, and miRNA regulation. This underscores THC’s potential therapeutic efficacy in modulating immune responses and mitigating inflammatory conditions.

An additional finding of this analysis was the difference between cell marker alterations in the duodenum of female macaques compared with male macaques, as demonstrated by the results of two of the studies [38,48]. Although the sample sizes, particularly for females, were too small to draw strong conclusions, this evidence aligns with the growing body of research on sex differences in the effects of THC and cannabinoids [114,115,116]. This finding underscores the importance of considering sex as a biological variable in future research. Additionally, as the conflicting results were observed in the duodenum of both sexes and not in blood, these findings further suggest the tissue-specific anti-inflammatory effects of THC. Emerging evidence also supports this hypothesis. In a mouse model of allergic contact dermatitis, topical THC showed anti-inflammatory activity independent of CBR1 and CBR2 receptors [117], indicating that THC may have different effects in different tissues due to the involvement of cell-type-specific receptors. Similarly, in another mouse model in BV-2 microglial cells, THC reduced pro-inflammatory cytokine production without involving CBR1, CBR2, or abn-CBD-sensitive receptors, while CBR2 receptor activation in LPS-stimulated MG-63 cells was partially involved in THC’s anti-inflammatory effects [118], again indicating the cell-type-specific and tissue-specific effects of THC.

It should be noted that previous studies have shown inconsistent results regarding THC’s effect on viral load. One study reported enhanced HIV replication following THC treatment [55], while another reported a decrease in virus replication [56]. Our analyses revealed that THC administration is not associated with significant changes in SIV replication in different tissues, especially in plasma and the colon, for which, due to the large sample size, we achieved good statistical power.

Overall, in this research, we reviewed evidence showing THC’s immunomodulatory and anti-inflammatory properties in SIV infection, via cell-type- and tissue-specific molecular mechanisms. Epigenetic mechanisms like miRNA regulation also influence THC’s impact on the transcriptome. These insights could help manage chronic inflammation in PWH.

### 4.2. Limitations in the Evidence

One major factor that should be considered is the stage of HIV infection, as T-cell responses vary at different stages. Early in the infection, there is typically an increase in the proportion of highly activated (CD38^+^) and proliferating (Ki-67^+^) CCR5^+^CD4^+^ T cells [119]. However, as the infection progresses, apoptosis and the cytopathicity of the virus lead to massive depletion of CD4^+^ T cells, especially in the gastrointestinal tract [120]. These factors should be taken into consideration when interpreting results from studies of the acute phase of the infection in contrast to studies from the chronic phase. Two of the included studies [39,51] were conducted within the initial 60 days of the subjects’ SIV infection. This could potentially have been a contributing reason for the study by Chandra et al., 2015 [39] not finding any significant difference in the expression of pro-inflammatory genes between THC+ and THC− subjects, as the host anti-inflammatory responses are also highly active during this stage of the infection. Similarly, Lyu et al. [93] reported that THC significantly reduced BEV concentration at 30 days but not at 150 days post-infection, suggesting that THC’s effects may be more pronounced in the acute phase of SIV infection. This temporal variation reinforces the need to consider the stage of infection when evaluating THC’s immunomodulatory impact. Other research gaps include biases due to inadequate statistical methods and unmatched treatment-group characteristics.

### 4.3. Limitations in Review Processes

Some limitations exist in this review. Firstly, the findings need to be interpreted with clinical, histopathological, and molecular evidence. For example, THC extended survival and reduced inflammation-driven lymph node fibrosis in SIV-infected macaques. Secondly, variability across tissues limited the feasibility of pooled analyses, and some studies included insufficient sample sizes. Furthermore, there was no quantitative meta-analysis on gene and miRNA expression, due to a lack of study data. Moreover, some challenges are associated with orthologous miRNA substitutions, due to evolutionary divergence and species-specific biology. Also, our analysis was not tissue-specific. Finally, due to the nature of the extracted data, it was not feasible to assess heterogeneity or the risk of non-reporting bias.

### 4.4. Implications

Future studies might use Bayesian statistics or hierarchical models to improve the current findings. Considering the high costs of studies in non-human primates (NHPs), human postmortem brain studies and in vitro models such as induced pluripotent stem cells (iPSCs) are being used to study the effects of cannabinoids. It is recommended that advanced single-cell transcriptomic and epigenomic technologies be integrated to deepen the understanding of THC’s effects. For instance, the Single-Cell Opioid Responses in the Context of HIV (SCORCH) U01 program [121], funded by the National Institute on Drug Abuse (NIDA), supports investigations into how substances such as opioids, methamphetamine, cocaine, and cannabinoids affect brain function, particularly in individuals with persistent HIV infection, using cutting-edge techniques like single-cell RNA sequencing to analyze individual brain cells. Translating NHP findings to clinical studies among PWH is crucial, with ongoing trials (NCT04800159, NCT05554146) offering broader perspectives. 

Additionally, this review focuses solely on THC. The effects of other cannabinoids, such as CBD, warrant evaluation in future reviews. For instance, a study by Arias et al. [122] demonstrated that CBD reduced levels of inflammatory cytokines, including IL-6 and IL-8, and decreased HIV expression in infected human microglial cells. These effects were achieved through the deactivation of caspase 1 and a reduction in *NLRP3* expression. These findings suggest that other cannabinoids, like CBD, may offer anti-inflammatory potential in the context of HIV, justifying further exploration.

## Figures and Tables

**Figure 1 ijms-26-02598-f001:**
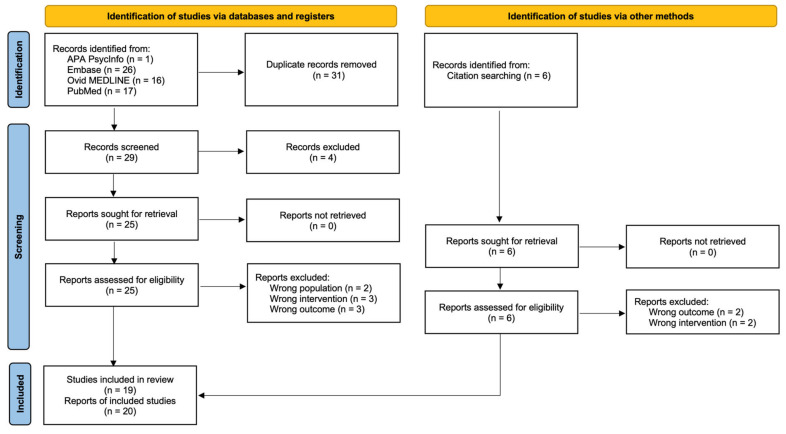
PRISMA flow diagram of the study selection process.

**Figure 2 ijms-26-02598-f002:**
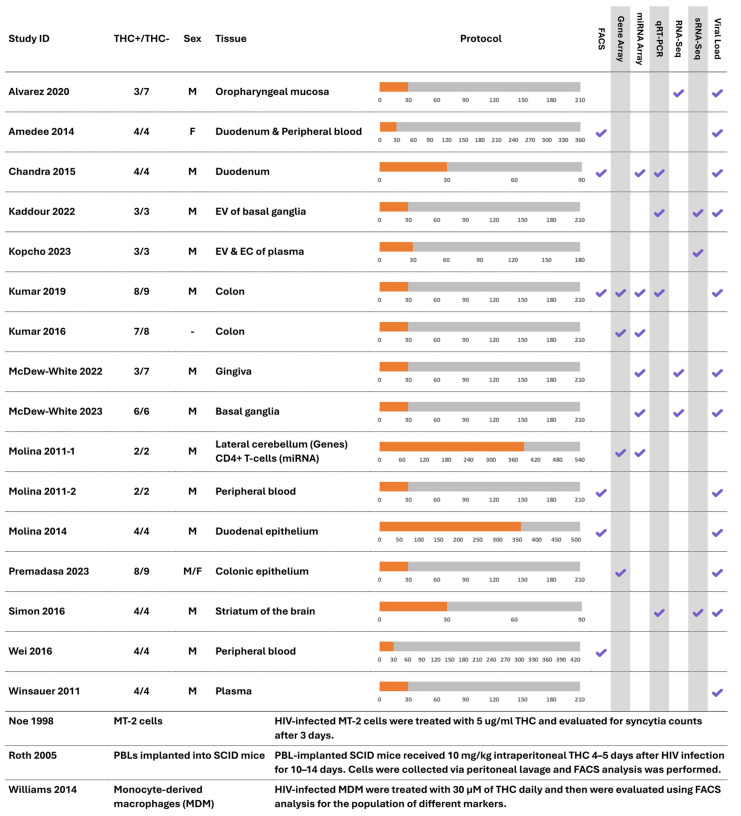
The characteristics of the studies included in the analyses [37,38,39,40,41,42,43,44,45,46,47,48,49,51,52,53,54,55,56]. The orange color of the protocol bar indicates the duration of pre-SIV infection treatment (THC or vehicle), while the grey color indicates the duration of post-SIV infection treatment (THC or vehicle) in the subjects. The numbers under each protocol bar indicate the number of days in each trial. EC: extracellular condensates; EV: extracellular vesicles; FACS: fluorescence-activated cell sorting; MDM: monocyte-derived macrophages; miRNA: micro-RNA; PBL: peripheral blood lymphocyte; qRT-PCR: quantitative reverse transcriptase polymerase chain reaction; RNA-Seq: RNA sequencing; SCID: severe combined immunodeficiency; sRNA-Seq: small RNA sequencing.

**Figure 3 ijms-26-02598-f003:**
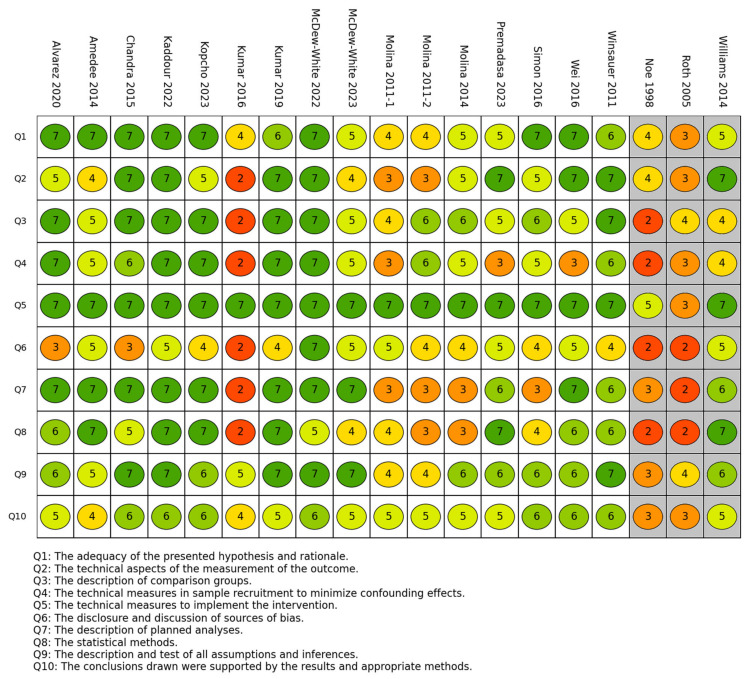
The methodological and reporting quality of the studies included in this review [37,38,39,40,41,42,43,44,45,46,47,48,49,50,51,52,53,54,55,56]. A score of 7 (dark green) indicates the highest quality, while 1 indicates the lowest (dark red). The three columns highlighted in grey represent studies that utilized human cells.

**Figure 4 ijms-26-02598-f004:**
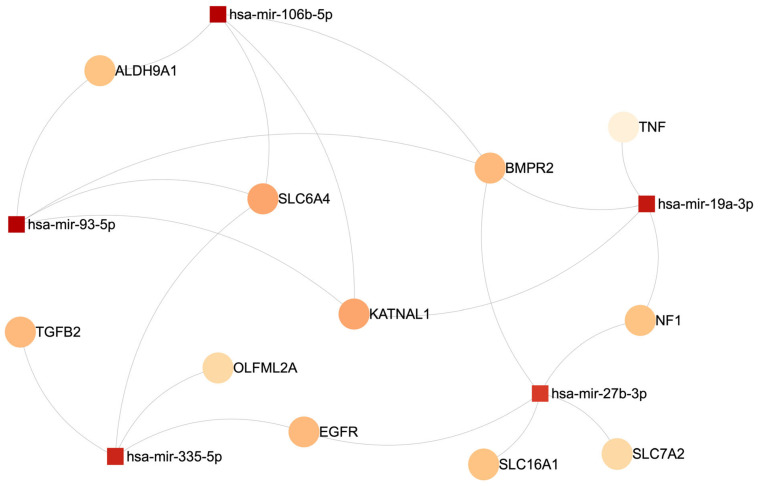
Top 5 differentially expressed miRNAs with the most connections to genes both differentially expressed and predicted as miRNA targets.

**Figure 5 ijms-26-02598-f005:**
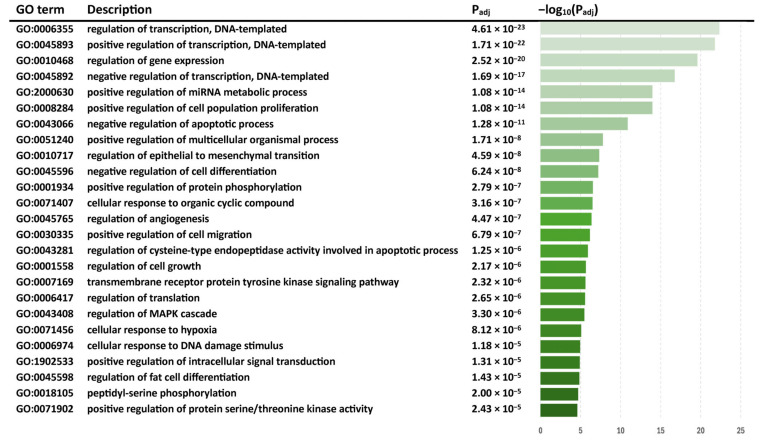
Top Gene Ontology Biological Process (GO BP) terms of differentially expressed miRNAs in SIV+/THC+ subjects compared with SIV+/THC− subjects in the included studies.

**Table 1 ijms-26-02598-t001:** Some differentially expressed genes found to be altered by THC in SIV-infected macaques that are relevant to HIV-associated inflammatory complications.

Study ID	Target Tissue	Finding
Alvarez 2020 [37]	Oropharyngeal mucosa	**↓** Pro-inflammatory genes: *SAMHD1*, *PELI3, KLK6*, and *ICAM1* **↓** Anti-inflammatory regulatory genes: *SOCS3*, *NR1D1*, and *IL4R*
Chandra 2015 [39]	Duodenum	**~** Pro-inflammatory cytokine genes: *TNF*, *IL1B*, *CCL2*, *CXCL11*, and *IFNG*
Kumar 2016 [42]	Colon	**↓** Pro-inflammatory genes: *DEFA4*, *DEFA5*, *DEFA6*, and *MMP8*
Kumar 2019 [43]	Colon	**↑** Tight junction protein genes: *CLDN3* and *OCLN* **↑** *PROM1* (an intestinal epithelial regeneration-associated gene) **↑** *MUC13* (an inhibitor of intestinal inflammation and epithelial apoptosis)
McDew-White 2022 [44]	Gingiva	**↓** Pro-inflammatory genes: *ADA2*, *ALOX5AP*, *CSF3R*, *IL21R*, *CXCL10*, *CXCR6*, *GADD4B*, *KLK6*, *KLRD1*, *LTB*, *MAMU-DRB1*, *SEMA7A*, *SLA*, *TIMP1*, and *VCAM1* **↓** Anti-inflammatory regulatory genes: *IDO1*, *MRC1*, *STAB1*, and *TNFAIP6* **↑** Oral epithelial barrier regulating genes: *CASP14*, *CDH13*, *DSC3*, *ITGA3*, *ITGA6*, *KRT10*, and *LAMB4* **↑** Active anti-inflammatory genes: *PARD3B*, *TRIM35*, and *TGFB2* **↑** Tissue protection and repair genes: *HP* and *PDGFC*
McDew-White 2023 [45]	Basal ganglia	**↑** Neuropeptide signaling pathway genes: *GLRA2*, *TAC3*, *CARTPT*, and *GRP* **↓** Cytoskeleton organization genes: *KRT4*, *KRT5*, *KRT6A*, *KRT15*, *KRT16*, *KRT23*, *KRT24*, *KRT76*, *KRT78*, and *KRT80*
Molina 2011-1 [46]	Lateral cerebellum tissue	**↓** *IKBKG* and *CARM1* (pro-inflammatory genes)
Premadasa 2023 [49]	Colonic epithelium	**↓***CCL15* (associated with neutrophil infiltration) **↑** *TIMP2* (a tissue protection and repair gene)
Simon 2016 [51]	Striatum of the brain	**↑***BDNF* (an important modulator gene of neuronal survival) **↓** *TNF* (a pro-inflammatory cytokine gene)

Note: The arrows show the relative expression levels in SIV+/THC+ compared with SIV+/THC− macaques. ↑ indicates higher expression levels, ↓ indicates lower expression levels, and ~ indicates no significant difference in expression between the groups.

**Table 2 ijms-26-02598-t002:** Significant THC-associated Gene Ontology Biological Process (GO BP) terms in SIV-infected macaques.

Term ID	GO BP Terms	P	P_adj_	Overlap	OR
GO:0045109	intermediate filament organization	2.43 × 10^−10^	4.82 × 10^−7^	12/68	15.1
GO:0014066	regulation of phosphatidylinositol 3-kinase signaling	7.24 × 10^−7^	4.78 × 10^−4^	10/90	8.8
GO:0098742	cell–cell adhesion via plasma-membrane adhesion molecules	1.42 × 10^−6^	7.07 × 10^−4^	13/172	5.8
GO:1901701	cellular response to oxygen-containing compound	2.19 × 10^−6^	8.70 × 10^−4^	20/406	3.7
GO:0060429	epithelium development	2.64 × 10^−6^	8.73 × 10^−4^	12/154	5.9
GO:0071677	positive regulation of mononuclear cell migration	4.78 × 10^−6^	1.05 × 10^−3^	6/31	16.6
GO:0090026	positive regulation of monocyte chemotaxis	4.54 × 10^−6^	1.05 × 10^−3^	5/18	26.6
GO:1902533	positive regulation of intracellular signal transduction	9.23 × 10^−6^	1.66 × 10^−3^	22/525	3.1
GO:0071222	cellular response to lipopolysaccharide	1.34 × 10^−5^	2.21 × 10^−3^	10/124	6.1
GO:0010605	negative regulation of macromolecule metabolic process	1.82 × 10^−5^	2.58 × 10^−3^	12/186	4.8
GO:0019221	cytokine-mediated signaling pathway	2.50 × 10^−5^	3.02 × 10^−3^	14/257	4.1
GO:0016525	negative regulation of angiogenesis	3.53 × 10^−5^	3.51 × 10^−3^	8/86	7.1
GO:0051241	negative regulation of multicellular organismal process	3.48 × 10^−5^	3.51 × 10^−3^	13/231	4.2
GO:0043405	regulation of MAP kinase activity	4.29 × 10^−5^	4.05 × 10^−3^	9/114	6.0
GO:0050727	regulation of inflammatory response	5.16 × 10^−5^	4.27 × 10^−3^	13/240	4.0
GO:0043588	skin development	5.70 × 10^−5^	4.45 × 10^−3^	7/68	8.0
GO:0061626	pharyngeal arch artery morphogenesis	5.84 × 10^−5^	4.45 × 10^−3^	3/6	68.7
GO:0010604	positive regulation of macromolecule metabolic process	8.96 × 10^−5^	6.58 × 10^−3^	16/364	3.2
GO:0046425	regulation of receptor signaling pathway via JAK-STAT	1.14 × 10^−4^	7.81 × 10^−3^	6/53	8.8
GO:0048660	regulation of smooth muscle cell proliferation	1.27 × 10^−4^	7.81 × 10^−3^	6/54	8.7
GO:0042417	dopamine metabolic process	1.41 × 10^−4^	8.00 × 10^−3^	4/19	18.4
GO:0043280	positive regulation of cysteine endopeptidase in apoptosis	1.90 × 10^−4^	9.92 × 10^−3^	8/109	5.5
GO:2001235	positive regulation of apoptotic signaling pathway	2.74 × 10^−4^	1.27 × 10^−2^	6/62	7.4
GO:0045670	regulation of osteoclast differentiation	3.39 × 10^−4^	1.40 × 10^−2^	5/42	9.3
GO:0001934	positive regulation of protein phosphorylation	4.30 × 10^−4^	1.55 × 10^−2^	15/377	2.9
GO:0006865	amino acid transport	5.21 × 10^−4^	1.77 × 10^−2^	5/46	8.4
GO:0033275	actin-myosin filament sliding	6.02 × 10^−4^	1.84 × 10^−2^	3/12	22.9
GO:0045429	positive regulation of nitric oxide biosynthetic process	5.84 × 10^−4^	1.84 × 10^−2^	4/27	12.0
GO:0051260	protein homo-oligomerization	5.94 × 10^−4^	1.84 × 10^−2^	8/129	4.6
GO:1903428	positive regulation of reactive oxygen species biosynthesis	6.02 × 10^−4^	1.84 × 10^−2^	3/12	22.9
GO:0071248	cellular response to metal ion	8.02 × 10^−4^	2.18 × 10^−2^	8/135	4.4
GO:0046942	carboxylic acid transport	1.00 × 10^−3^	2.46 × 10^−2^	5/53	7.2
GO:0051091	positive regulation of DNA-binding transcription factor activity	9.73 × 10^−4^	2.46 × 10^−2^	11/246	3.3
GO:0030198	extracellular matrix organization	1.10 × 10^−3^	2.67 × 10^−2^	9/176	3.7
GO:0002719	negative regulation of cytokine production in immunity	1.46 × 10^−3^	3.27 × 10^−2^	3/16	15.8
GO:1904646	cellular response to amyloid-beta	1.42 × 10^−3^	3.27 × 10^−2^	4/34	9.2
GO:0001755	neural crest cell migration	1.77 × 10^−3^	3.69 × 10^−2^	4/36	8.6
GO:0050772	positive regulation of axonogenesis	2.16 × 10^−3^	4.05 × 10^−2^	4/38	8.1
GO:0042509	regulation of tyrosine phosphorylation of STAT protein	2.51 × 10^−3^	4.49 × 10^−2^	5/65	5.7
GO:0032880	regulation of protein localization	2.89 × 10^−3^	4.95 × 10^−2^	6/97	4.6
GO:0051045	negative regulation of membrane protein proteolysis	3.02 × 10^−3^	4.99 × 10^−2^	2/6	34.2

Note: BP: biological process; GO: gene ontology; JAK: Janus kinase; MAP: mitogen-activated protein; OR: odds ratio; P_adj_: adjusted *p* value; STAT: signal transducer and activator of transcription protein.

**Table 3 ijms-26-02598-t003:** Some differentially expressed miRNAs found to be altered by THC in SIV-infected macaques that are relevant to HIV-associated inflammatory complications.

Study ID	Target Tissue	Results
Chandra 2015 [39]	Duodenum	**↑** Immunomodulatory miRNAs: miR-149, miR-24, and miR-99
Kaddour 2022 [40]	Extracellular vesicles of basal ganglia	**↑** Neuroinflammation regulatory miRNAs: mml-let-7a-5p and mml-let-7c-5p
Kopcho 2023 [41]	Extracellular vesicles and condensates from plasma	**↓** miR-335-5p (a regulator of the PI3K-Akt pathway) **↓** miR-139-5p (a regulator of the MAPK signaling pathway)
Kumar 2016 [42]	Colon	**↑** Immunomodulatory miRNAs: miR-193b-5p and miR-374a-5p
Kumar 2019 [43]	Colon	**↓** Pro-inflammatory miRNAs: miR-21, miR-141, and miR-222
McDew-White 2022 [44]	Gingiva	**↓** Pro-inflammatory miRNAs: miR-142-3p, miR-223, miR-146a, and miR-34c
McDew-White 2023 [45]	Basal ganglia	**↑** miR-218-5p (a neuroinflammation regulatory miRNA)
Molina 2011-1 [46]	CD4^+^ cells	**↑** miRNAs associated with the regulation of T-cell activation: miR-142-3p, miR-142-5p, and miR-150
Simon 2016 [51]	Striatum of the brain	**↑** miR-105-5p (associated with nervous system development) **↑** miR-767-5p (an immunomodulatory miRNA)

Note: The arrows show the relative expression levels in SIV+/THC+ compared with SIV+/THC− macaques. ↑ indicates higher expression levels, and ↓ indicates lower expression levels.

**Table 4 ijms-26-02598-t004:** Levels of immune cell markers in SIV+/THC+ compared with SIV+/THC− subjects.

Cell Type	Duodenum				Peripheral Blood				
	Amedee 2014 [38]	Chandra 2015 [39]	Kumar 2019 [43]	Molina 2014 [47]	Amedee 2014 [38]	Kumar 2019 [43]	Molina 2011-2 [47]	Molina 2014 [47]	Wei 2016 [52]
B-cell	-	-	-	-	-	-	-	-	~
B-cell IgE^+^	-	-	-	-	-	-	-	-	~
Macrophage (CD3^−^CD14^−^)	-	-	↑	-	-	-	-	-	-
T-cell CD4^+^	~	↑	~	~	-	~	~	~	~
T-cell CD4^+^ (apoptosis marker)	-	-	-	-	-	-	~	-	-
T-cell CD4^+^ (CCR5+ total)	~	-	-	~	~	-	-	~	-
T-cell CD4^+^ (central)	↑	-	-	~	~	-	-	~	-
T-cell CD4^+^ (CXCR4^+^ total)	~	-	-	~	~	-	-	~	-
T-cell CD4^+^ (effector)	↑	-	-	~	~	-	-	~	-
T-cell CD4^+^ (programmed death marker)	-	-	↓	-	-	-	-	-	-
T-cell CD4^+^ (proliferation marker)	-	-	~	-	-	-	~	-	~
T-cell CD8^+^	~	~	~	~	-	~	~	~	~
T-cell CD8^+^ (activation marker)	-	-	↓	-	-	-	-	-	-
T-cell CD8^+^ (apoptosis marker)	-	-	-	-	-	-	~	-	-
T-cell CD8^+^ (CCR5^+^)	~	-	-	~	~	-	-	~	-
T-cell CD8^+^ (Central)	~	-	-	↑	↓	-	-	~	-
T-cell CD8^+^ (CXCR4^+^)	~	-	-	~	~	-	-	~	-
T-cell CD8^+^ (effector)	~	-	-	~	~	-	-	~	-
T-cell CD8^+^ (programmed death marker)	-	-	↓	-	-	-	-	-	-
T-cell CD8^+^ (proliferation marker)	-	-	~	-	-	-	~	-	~

Note: The arrows show the relative expression levels in SIV+/THC+ compared with SIV+/THC− macaques. ↑ (green) indicates higher expression levels in SIV+/THC+ compared with SIV+/THC−, ↓ (orange) indicates lower expression levels in SIV+/THC+ compared with SIV+/THC−, and ~ (yellow) indicates no significant difference in expression between the groups.

## Data Availability

Data is contained within the article and Appendix A.

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
