# Peer review of "Transcriptomic Alterations Induced by Tetrahydrocannabinol in SIV/HIV Infection: A Systematic Review"

_ijms, 2025, doi:10.3390/ijms26062598_

Round 1

Reviewer 1 Report

Comments and Suggestions for Authors

Valizadeh et al in this review manuscript entitled “Transcriptomic alterations induced by tetrahydrocannabinol in HIV infection: A systematic review” have evaluated published studies in Simian Immunodeficiency Virus (SIV) infected macaques and a few HIV-infected human cells to show if Tetrahydrocannabinol has any possible role in modulation of the host transcriptome. The study was performed systematically, and the findings are interesting. However, I will suggest the authors to incorporate the followings points in the final version.

  1. With only three studies on HIV-1 infected human cells reported, and the published data reanalyzed come from mainly SIV infected macaques, I will suggest authors rethink about the manuscript title.
  2. In page 2, authors stated “Similar patterns were found with reports of past year (44.9% versus 24.4%) and past month (35.6% versus 17.2%) marijuana use among PWH compared to uninfected controls” . I will suggest the authors to indicate the exact time frame.
  3. References 7 and 8 are hard to retrieve. Are they correctly written?
  4. Authors are not consistent in writing CBR1 and CBR2. Is this CBR1 or CBR1…..?
  5. In page 4, 8. Effect measures, authors stated” the effect was measured based on the direction of the expression in the case group compared to the control group” . It looks like that authors compared the expression of genes in the case group compared to the control groups, but the word “direction” is misleading.
  6. In page 4, 9. Synthesis methods, authors have written “This approach entailed the replacement of a miRNA from one species-specific kit with a corresponding miRNA from another species-specific kit”. Is this a kit or some software? Any citations available should be provided.
  7. Figure 2, citations are not correctly written for example Noe 1998 is listed at 47 in the reference section. Also, what do the numbers below the protocol charts indicate?
  8. In page 9, authors stated” Out of the 15 studies on macaques, eight conducted evaluated differential miRNA expression analysis”. Use either conducted or evaluated.
  9. In table 4, what do the colors of rows indicate?
  10. Table 5: is this pooled analysis with such spread data worth to for any conclusion? My suggestion is to remove this table.
  11. In page 12, authors wrote “compared to that in control cells treated with VEH after 3 days”. Could the authors expand what is VEH?
  12. In page 13, authors stated “….exposure to THC can suppress immune function, increase HIV coreceptor expression, and act as a cofactor….”. This statement does not make sense as Roth et al used a CXCR4 tropic virus for infection and the expression of CXCR4 was not altered upon THC treatment.
  13. In page 14, authors wrote “For instance, miRNAs target DEGs encoding cytokines, receptors, and signaling molecules within the TNF and TGF-β pathways, affecting immune cell activation and cytokine production (75, 76). Additionally, these miRNAs regulate DEGs involved in cell adhesion, migration, and proliferation, such as ITGA3, EGFR…”. The term DEG should only be used when in comparison. Here in this context, instead of DEG, it should be only gene.
  14. In page 15, could the authors expand what is SCORCH U01 and add citation?
  15. Authors need to carefully go through the manuscript to fix grammatical errors.

Reviewer 2 Report

Comments and Suggestions for Authors

The article by Valizadeh and colleagues is a systematic review of the literature encompassing immunomodulatory effect of THC on SIV-infected macaques (15 studies). The authors summarize the previously reported findings on cellular activation in various anatomical localizations, but the focus is changes in gene expression and immune signaling pathways. An additional 3 in vitro studies using human cells are included.

This review is relevant to gain a better understanding of the effect of cannabis use on chronic inflammation during HIV infection, and the findings are of interest to people living with HIV, physicians and scientists working in the field of HIV and perhaps also to people in the field of other chronic inflammatory pathologies.

Overall, the review is well written but I have some concerns i) regarding the methodology, ii) the description of the translatability of macaque and human in vitro models to the human in vivo setting, and iii) a description of how THC can affect gene expression at the molecular level is missing, and this would be beneficial to understand how THC could be used as immunosuppressant.

Methodology:

The pubmed search was performed in November 2023. This seems a bit outdated and could be updated.

The criteria of inclusions and exclusion are not well described.

I think this article would be relevant to include for the SIV-inf macaque section: PMID: 33036231

It is not clear what the inclusion criteria are for the articles that report on in vitro studies with human cells and/or mouse/rat models with transplanted human cells. I think these studies could also be included in the review.

  • PMID: 37147420 – 2023 – where CBD and THC are compared for their anti-inflammatory effect in human microglial cells infected with HIV.
  • PMID: 24742657 – 2014 – Effect of THC on HIV Tat protein-induced U937 cell adhesion, mimicking transmigration of human monocytes across the blood brain barrier.
  • PMID: 28692581 – 2017 – THC inhibits IFNa production by pDCs from controls and people living with HIV.
  • PMID: 26733986 – 2015 – Effect of alcohol and THC on HIV infection of monocyte-derived DC and their function.
  • PMID: 30026298 – 2018 – THC inhibits IFNa-induced activation of T cells from controls and people living with HIV.
  • PMID: 34338765 – 2021 – HIV transgenic rats treated with THC

Translatability to human setting.

In the abstract/introduction the authors write that THC may modulate immune responses to HIV and this may aid people living with HIV, but I miss a paragraph where the findings from the macaques are evaluated for their relevance in the human setting. There are multiple reviews on the effect of cannabinoids and inflammation in people living with HIV (for example PMID: 34452386 and PMID: 31764093, also referenced by the authors) and I think it would bring the findings from this review to a higher scientific level if they were at least discussed in relation to the findings from the  human studies.

Molecular mechanisms

I miss a paragraph describing the molecular signal transduction route downstream the receptor for THC driving alteration in gene expression, for example the genes that are found to be suppressed by THC such as TNF and perhaps from the human in vitro studies the effect on HIV transcription. Currently the review gives a broad overview of differential immunological pathway regulation induced by THC but it does not give an understanding as to how THC does this at a molecular level.

Other points:

Page 1:

  • Title: “Transcriptomic alterations induced by tetrahydrocannabinol in HIV infection: A systematic review”. Would be more appropriate to reflect the SIV studies as they dominate the review.
  • “This review systematically evaluates preclinical studies in Simian Immunodeficiency Virus (SIV) infected macaques and HIV-infected human cells to elucidate how Tetrahydrocannabinol (THC), a principal compound of cannabis, may alleviate HIV symptoms by modulating the host transcriptome in response to HIV infection. “ Perhaps better rephrased to: THC-induced immunosuppression may alleviate chronic immune activation caused by HIV infection?
  • “THC may modulate inflammation and immune responses in HIV…”. Instead of “in HIV”, better “during HIV infection”.
  • “The emergence of the human immunodeficiency virus (HIV) in 1981 marked the onset of a persistent global health challenge.” This is incorrect. HIV emerged in the 1920s in the DRC, the 1980s mark the spread in Europe and North America.

Page 2:

  • “During the era of antiretroviral therapy (ART), a significant proportion of people with HIV (PWH) exhibit non-AIDS comorbidities. These conditions arise from sustained chronic immune activation and inflammation, encompassing complications such as atherosclerosis, liver fibrosis, and neuroinflammation (2).” This would benefit from 1-2 sentences highlighting the success of ART, such as HIV suppression to undetectable levels in plasma and prevention of transmission, together with the adverse effects from some individuals from either living with HIV or ART, which are difficult to separate.
  • “Previous studies on sooty mangabeys and African green monkeys infected with simian immunodeficiency virus (SIV), a virus closely associated with HIV-1, demonstrate that chronic inflammation, rather than viral replication, is likely the primary driver of AIDS progression.” Yes, this may be true for sooty mangabeys and AGM but that does not mean it applies to humans’ progression to AIDS as a consequence of infection with HIV. This sentence should be rephrased to reflect that it may contribute to progression to AIDS based on these two animal models but should also include other examples of animal models, preferably monkeys or great apes, that progress to AIDS upon infection with SIV or SHIV.
  • “Therefore, immunomodulatory agents such as cannabis may relieve symptoms in PWH or slow the progression to AIDS due to their ability to suppress inflammation (5).” Cannabis may give relieve of symptoms in some individuals but progression to AIDS is stopped by ART in all people, perhaps emphasize here that the use of cannabis is not to replace ART.

Page 3:

  • “SIVs have been extensively utilized to investigate HIV and AIDS using animal models.” Better rephrase, unclear what SIVs are.

Page 6:

  • Figure 2. references in the figure are incorrect.
  • Figure 2 legend: “The grey color of the chart indicates the duration of pre-infection THC treatment, while the orange color indicates the duration of post-infection THC treatment.” I think this is the opposite.
  • “Almost all macaque studies used the SIVmac251 strain, with the only exceptions being one SIVmac239 subject inoculated in two studies (32, 37) and all vehicle-untreated subjects inoculated with SIVmac239 in one study (38).” Unclear sentence, please rephrase.

Page 7:

  • SOCS3 (Suppressor of cytokine signaling 3) is described as a proinflammatory gene but it inhibits STAT signaling and negatively affects cytokine signaling. It is indeed induced as a result of inflammation, but the function is to suppress the inflammation via a negative feedback loop. As such, I would not group this gene in the list of pro-inflammatory genes.
  • Similar comment for

Page 11:

  • Figure 4. Not readable.
  • Table 4. The legend at the top is not easy to read.

Page 12:

  • “Only three studies have investigated alterations due to THC in HIV-infected humans (47-49).” This is incorrect. The studies investigated the effect of THC on human cells cultured in vitro or transplanted into a mouse, not the effect of THC in a human.
  • Description of the study by Noel et al. would benefit from explaining which type of cells MT2s are.

Comments on the Quality of English Language

The manuscript is well written. There are some sentences that benefit from rephrasing.

Round 2

Reviewer 2 Report

Comments and Suggestions for Authors

The authors have addressed my concerns and I think this improved the quality of the manuscript. I have no further comments.

Author Response

We are glad to see the reviewer's concerns have been addressed. Thank you for your helpful comments.